Review Article

**EMBO** *reports*

# X chromosome inactivation in mammals: general principles and species-specific considerations

Charbel Alfeghaly [1,2] & Claire Rougeulle [1,2✉]

## Abstract

**X chromosome inactivation (XCI) is a mammalian dosage compensation mechanism that ensures balanced expression of X-linked genes between males and females. Research using rodent models has led to major discoveries regarding XCI mechanisms and dynamics, in addition to the molecular actors involved in this process, including the long noncoding RNA *Xist* and its protein partners. However, several features of XCI vary significantly across mammalian species, not only between marsupials and placental mammals, but also within the latter. This review discusses the fundamental aspects of XCI from an evolutionary perspective, highlighting both conserved features and species-specific variations across mammalian species.**

**Keywords** X Chromosome Inactivation; Dosage Compensation; Long Noncoding RNAs; *Xist*; *RSX*
**Subject Categories** Chromatin, Transcription & Genomics; Evolution & Ecology; RNA Biology

## Introduction

Therian mammals, comprising marsupials and eutherians, exhibit heteromorphic sex chromosomes, with females having two X chromosomes (XX) and males having one X and a Y chromosome (XY). These sex chromosomes originated around 150 million years ago from a pair of autosomes, called proto-X and proto-Y.

It is widely hypothesized that the differentiation of sex chromosomes began with a mutation on the proto-Y, which led to the emergence of a male sex-determining gene known as *SRY* (Graves, 2006). Over time, the proto-Y and proto-X diverged markedly, likely due to suppression of recombination between the two former homologs, leading to X and Y chromosomes with radically different genetic content and structure. This process resulted in a two-level disequilibrium: an imbalance in X-linked gene dosage relative to autosomes in males (1X/2A) and a disparity in X-linked gene dosage between males (1X) and females (2X). A two-fold increase in the expression of X-linked genes has been proposed to resolve the X/A dosage, while the male/female X

chromosome disequilibrium is compensated for by the transcriptional silencing of one of the two X chromosomes in female mammals through the process of X chromosome inactivation (XCI).

The ins and outs of this fascinating process of concerted gene regulation have been mainly explored in an easily manipulated animal model, mice. This led to major discoveries regarding, among others, the timing of XCI onset in early female embryonic development, its absolute requirement for embryo viability, and the molecular actors at play, starting from the central role of the X-encoded long noncoding RNA (lncRNA) *Xist* to the coding and noncoding regulators of *Xist* expression, and to *Xist* protein partners (Sahakyan et al, 2018). XCI results in the inactivation of most, but not all X-linked genes carried by the *Xist*-coated chromosome, which is accompanied, and to some extent triggered, by intensive chromatin remodeling.

Given its wide representation among mammalian species and its essential role in mouse development, it came as a surprise to realize that several parameters of the XCI process display some degree of variability from one species to another. The most striking example comes from the comparison between placental mammals and marsupials, where XCI is triggered by distinct lncRNAs, *Xist* and *RSX*, respectively. These lncRNAs have independent evolutionary origins, while being both X-linked. However, differences in XCI features and mechanisms also exist among eutherians. This review aims to underscore the evolutionary aspects of XCI in therians, emphasizing the variability of XCI features across species, and exploring the origin and implications of such variability.

### The exquisite diversity in the noncoding regulators of XCI

A shared feature of XCI across species is its reliance on noncoding RNAs containing repeat elements, which may act as functional domains. The nature and mode of actions of these elements, however, display notable variability across both long and short evolutionary timescales.

#### XIST and RSX lncRNAs

The central role of *Xist* lncRNA in triggering XCI has been extensively demonstrated in mice, and to a lesser degree in humans, but is believed to be conserved in all placental mammals. *Xist* includes six conserved functional domains, made up of tandem repeat sequences known as repeats A-F, which can vary in copy

[1]Epigenetics and Cell Fate, CNRS, Université Paris Cité, Paris, France. [2]Present address: DIG-Cancer, CNRS UMR3244, Sorbonne Université, PSL University, Université Paris Cité, Institut Curie, Centre de Recherche, Paris F-75248, France. ✉E-mail: claire.rougeulle@curie.fr

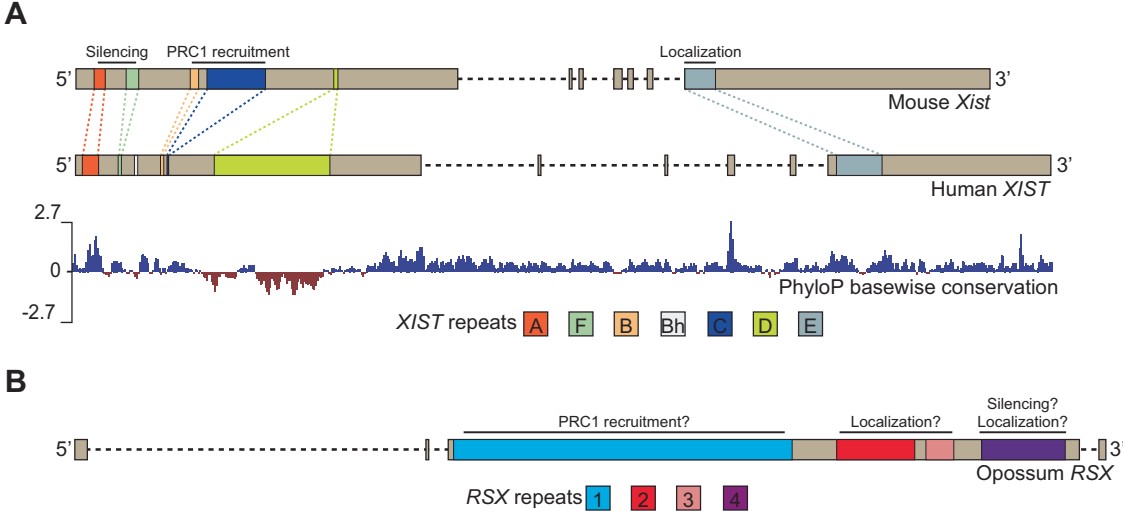

**Figure 1. Comparison of *Xist* and *RSX* lncRNAs' tandem repeats.**

(A) Gene structure and tandem repeat organization of mouse and human *Xist/XIST*. Exons are represented by brown boxes, and introns by lines. The colored boxes highlight different tandem repeat domains within the lncRNA gene. Specific functions related to each repeat domain, when known, are indicated. The PhyloP conservation track from UCSC represents evolutionary conservation across species and is based on the human *XIST* sequence. Positive values (in blue) indicate conserved nucleotides, while negative values (in red) indicate accelerated evolution. (B) Similar to (A) but showing the gene structure and tandem repeat organization of *RSX* in opossums.

number across different species (Fig. 1A). These repeat domains were shown, mainly in the mouse and to a lesser extent in humans, to be crucial for *Xist*'s functions, through interaction with specific RNA-binding proteins (RBPs) (Raposo et al, 2021). Repeat A is indispensable for silencing and is under selective constraints based on PhyloP scores, highlighting its crucial role in XCI (Fig. 1A) (Wutz et al, 2002). The requirement of the human *XIST* A repeat for silencing has been demonstrated in a transgenic context (Dixon-McDougall and Brown, 2022). The human F repeat was also shown to play a role in gene silencing in this system. This might be linked to its role in tethering the Xi to lamin B at the nuclear periphery (Chen et al, 2016). Repeats B and C are involved in the recruitment of polycomb proteins and the deposition of the repressive marks H2AK119Ub and H3K27me3 (Almeida et al, 2017; Pintacuda et al, 2017). In humans, the B repeat is duplicated, including an additional B-like repeat known as Bh, while the C repeat is significantly shorter compared to the 14 copies found in mice (Dixon-McDougall and Brown, 2021). Deleting the Bh repeat in humans reduced H3K27me3, indicating a similar role for the human repeat as in mice (Dixon-McDougall and Brown, 2021). Repeat E anchors *Xist* to the Xi but does not intervene in XCI initiation (Pandya-Jones et al, 2020). The function of repeat D is the least understood. In rodents, it is strongly degenerated compared with humans and other eutherians. Studies suggest a role for repeat D in regulating *XIST* expression in human HEK293 cells, but it does not appear to influence *Xist* expression, coating, or the deposition of repressive histone marks in mouse embryonic fibroblasts, consistent with its truncated nature (Colognori et al, 2019; Lv et al, 2016). Despite these differences at the nucleotide level, the higher-order structure of *Xist* RNA has been found to be conserved between mouse and human (Lu et al, 2020, 2016).

*Xist* evolved from the *Lnx3* protein-coding gene after the divergence from marsupials, implying that XCI in marsupials relies on a distinct set of factors (Duret et al, 2006; Hore et al, 2007;

Shevchenko et al, 2007). The presence of eutherian-like features on the Xi, such as H3K27me3, which is *Xist*-dependent in eutherians, suggested the presence of another lncRNA with *Xist*-like properties. RNA-FISH analysis in opossum led to the identification of *RSX* (RNA on the silent X), which displayed female-specific expression in different opossum tissues and was transcribed exclusively from the Xi (Grant et al, 2012). The insertion of an *RSX* transgene into mouse ESCs induced gene silencing in *cis*, further supporting its involvement in marsupial XCI (Grant et al, 2012).

The *RSX* gene is located in a different syntenic block than *Xist* and is conserved in both South American and Australasian metatherian groups. This suggests that *RSX* originated before this major split and is common among all Metatheria (Grant et al, 2012). Despite the distinct evolutionary origins and lack of sequence similarity between *Xist* and *RSX*, some features are shared. Notably, *RSX* contains long domains of four tandem repeats (repeats 1–4), which are hypothesized to function similarly to *Xist* repeats (Fig. 1B) (Sprague et al, 2019). K-mer-based analyses revealed that the repeat A of *Xist* showed the highest similarity to *RSX* repeat 4, while repeat B aligned most closely with *RSX* repeat 1. Human *XIST* repeat D positively correlated with *RSX* repeats 1 and 4, while *Xist* repeat E demonstrated correlations with *RSX* repeats 2, 3, and 4. In contrast, *Xist* repeat C exhibited no significant correlation with any *RSX* repeat, and *Xist* repeat F was not included in the analysis. These k-mer similarities align with domain-specific enrichment of protein-binding motifs, suggesting that both lncRNAs may recruit a similar set of protein partners.

## Commonalities and differences in Xist and RSX interactomes across species

The development of various RNA-centric methods for investigating RNA-protein interactions has enabled the comprehensive identification of the *Xist* interactome (Bousard et al, 2019; Chu et al, 2015; McHugh et al, 2015; Minajigi et al, 2015), providing an in-

depth examination of the mechanisms through which *Xist* silences the X chromosome. In total, 494 proteins were found to bind *Xist* RNA across multiple mouse cell lines and through various experimental strategies. Many of these proteins were known to have roles in chromatin modifications, the nuclear matrix, and RNA remodeling pathways (Chu et al, 2015; Minajigi et al, 2015). Although only a few proteins were commonly identified by these different studies, largely due to differences in methodologies and cell types, some consistently appeared across all studies. Among these factors, the transcriptional repressor SPEN (also known as SHARP in human and MINT in mouse) plays an essential role in *Xist*-mediated gene silencing (Chu et al, 2015; Dossin et al, 2020; McHugh et al, 2015; Moindrot et al, 2015; Monfort et al, 2015). SPEN interacts with the A repeat of *Xist* and recruits the corepressors NCoR/SMRT, which in turn activate histone deacetylase 3 (HDAC3). This drives the removal of acetylation marks from histones H3 and H4, a key step in the transcriptional silencing of the future Xi (Dossin et al, 2020; Żylicz et al, 2019). Additionally, members of the hnRNP family, such as hnRNP U and hnRNP K, were consistently identified. HnRNP U was previously known for its accumulation on the Xi and role in *Xist* localization (Hasegawa et al, 2010). HnRNP K acts as a scaffold protein. It binds to repeats B and C of *Xist* and recruits the noncanonical PRC1 complex to deposit H2AK119Ub (Bousard et al, 2019; Nakamoto et al, 2020; Pintacuda et al, 2017). The H2AK119Ub mark is then recognized by PRC2 via JARID2, triggering the deposition of H3K27me3 (Cooper et al, 2016). Both repressive histone modifications have been recently shown to be essential for maintenance of XCI (Masui et al, 2023). The *Xist* interactome is also enriched with proteins containing intrinsically disordered regions (IDRs), which are essential for mediating protein-protein and protein-RNA interactions as well as the formation of phase-separated condensates (Cerase et al, 2019; McIntyre et al, 2024; Zhao et al, 2021). Computational analysis predicted that over half (54%) of *Xist*-bound proteins are prone to phase separation, aligning with findings that *Xist* RNA forms large protein assemblies known as supramolecular complexes (SMACs) (Cerase et al, 2019). These complexes consist of hundreds to thousands of protein molecules organized into biophysically unique domains within the nucleus (Markaki et al, 2021; Pandya-Jones et al, 2020). The formation of SMACs depends on SPEN recruitment and appears essential for silencing genes at various distances from *Xist* RNA (Jachowicz et al, 2022; Markaki et al, 2021). For genes further away, this silencing relies on chromatin compaction, mediated by the PRC1 complex (Markaki et al, 2021).

Although most *Xist* pulldown experiments have been conducted in mice, the composition of human *XIST* interactors was recently characterized in two somatic immune cell types: GM12878 and K562 (Yu et al, 2021). Gene ontology analysis revealed that many pathways are conserved when mice and human *XIST* interactors are compared, despite only 57% of factors being identical, including key regulators of XCI such as SPEN and hnRNP K. Such discrepancy might be due to the different stages of XCI being studied: establishment in mice versus maintenance in humans. Of note, there is already evidence of cell-type-specific factors, at least during XCI maintenance. The comparison of *XIST* interactors between the two human somatic cell lines showed only 71% overlap. For example, TRIM28 was detected exclusively in GM12878 cells and not in K562 cells. The loss of TRIM28 led to

the reactivation of a subset of X-linked genes specifically in GM12878 cells, confirming its cell-type specificity (Yu et al, 2021). It is thus impossible to conclude whether there are species-specific or cell-type-specific differences in *XIST* interactors. A more appropriate comparison would involve examining *XIST* interactors during the establishment phase of XCI in human cells.

The recent identification of the *RSX* interactome has provided valuable insights into the molecular mechanisms underlying *RSX*'s role in silencing the marsupial X chromosome, raising important questions about how these mechanisms compare to those of *XIST*. McIntyre et al identified 135 proteins that bind to *RSX* in *Monodelphis* fibroblast cells (McIntyre et al, 2024). Gene set enrichment analysis revealed that over 90% of the ontology terms were commonly found in the *RSX* and mouse *Xist* interactomes, reinforcing the functional similarities of these lncRNAs despite their independent evolution. Notably, hnRNP K was identified as a common protein among the interactomes, in agreement with H2AK119Ub and H3K27me3 being hallmarks of the Xi in eutherians and marsupials (Chaumeil et al, 2011). On the contrary, only 40% of the *RSX* interactome shared orthologues with the *Xist* interactome. This disparity could again be due to differences in the cell types and XCI stages studied across species, or to experimental artifacts, which would require additional validation. However, it may also suggest the involvement of unique proteins in the XCI process specific to marsupials, potentially reflecting divergence in certain XCI mechanisms between eutherians and marsupials. One such *RSX*-specific interactor is CKAP4, which was originally described to have a major function in organizing the overall structure of the endoplasmic reticulum (Tuffy and Planey, 2012). Depletion of CKAP4 increased the proportion of cells with biallelic expression of the X-linked gene MSN, pointing to a potential role in marsupial XCI, which warrants further investigation.

Despite the low overlap of common interactors with *Xist*, *RSX*'s interactome is enriched in proteins with IDRs, suggesting that *RSX* may utilize similar mechanisms as *Xist*, involving non-stoichiometric recruitment of protein partners through their IDRs to achieve chromosome-wide gene silencing. It is worth noting that SPEN was not identified in the reported *RSX* interactome, even though a SPEN orthologue exists in marsupials. This raises the question of whether its absence is due to SPEN not being required at later stages of XCI or if another protein with similar functions mediates the formation of SMACs in marsupials. Future studies will reveal whether *RSX* forms SMACs, and what their composition and role in marsupial XCI are.

## Status of the inactive X chromosome

The differences outlined above regarding the noncoding regulators of XCI and their associated protein partners raise significant questions about the consequences for XCI outcomes across species.

### Commonalities and diversity in chromatin and structural features of the inactive X chromosome across therians

*Xist* and *RSX* interact with numerous factors that directly or indirectly modify the chromatin nature of the X chromosome being inactivated, in agreement with long-known XCI-associated chromatin reorganization (Żylicz and Heard, 2020). The Xi is generally depleted in histone modifications associated with active transcription and enriched in repressive histone modifications in both

eutherians and marsupials (Chaumeil et al, 2011; Rens et al, 2010). The nature and distribution of these modifications can vary across species and even among cell types within the same species (Vallot et al, 2016; Waters and Waters, 2021). Certain marks, however, such as H3K27me3 and H2AK119Ub, are consistently observed across studies and cell types, in accordance with the conserved interaction of hnRNP K with both *Xist* and *RSX*.

A key difference between the eutherian and marsupial Xi is DNA methylation of CpG islands, which is considered a late-stage modification in the XCI process, which helps maintain the repressed state of inactivated genes. This epigenetic modification depends on the de novo methyltransferase activity of DNMT3B and on *Xist*-mediated recruitment of the SMCHD1 protein to the Xi (Gendrel et al, 2012; Yagi et al, 2020). Studies in eutherians have shown that intergenic and promoter regions of silenced genes on the Xi are generally methylated (Balaton and Brown, 2021; Cotton et al, 2015, 2011; Sharp et al, 2011), whereas the opposite has been found in marsupials: transcription start sites of X-linked genes remain largely unmethylated in all species analyzed, including opossums and koalas (Singh et al, 2021; Wang et al, 2014; Waters et al, 2018). The causes of this variability and its consequences on gene silencing remain poorly understood. However, the absence of SMCHD1 in the *RSX* interactome may be potentially linked to the lack of Xi methylation in marsupials.

In addition to the aforementioned chromatin modifications, the Xi undergoes significant three-dimensional remodeling during XCI. Although this has been exclusively analyzed in eutherians, studies across multiple species and tissues revealed a unique Xi configuration, characterized by the absence of topologically associated domains (TADs) and the formation of two large mega-domains, separated by the DXZ4 boundary region (Bauer et al, 2021; Collombet et al, 2020; Giorgetti et al, 2016; Wang et al, 2018). The functional importance of this region has been thoroughly investigated in humans and mice, where deletion of DXZ4 disrupts the bipartite structure of the Xi without impacting the initiation or maintenance of XCI, suggesting that DXZ4 and the associated mega-domains may not play a crucial role in XCI (Andergassen et al, 2019; Bonora et al, 2018; Darrow et al, 2016; Fang et al, 2020; Froberg et al, 2018; Giorgetti et al, 2016). A recent study revealed that the DXZ4 boundary region is conserved in elephants and could also be detected in mammoths through paleo HiC, but the Xi in these species displays a tetradic structure with four mega-domains (Sandoval-Velasco et al, 2024). This distinctive organization includes two additional boundary elements, one of which is also found in humans but does not act as a boundary there. These findings highlight for the first time a remarkable diversity in the Xi structural organization. Expanding research to include marsupials and other eutherian species will be essential for understanding how the 3D organization of the Xi has evolved across mammals.

### Not all inactive Xs are equally silent: the peculiar case of escapees

The functional outcome of these chromatin remodeling events is stable transcriptional silencing of the majority of genes on the Xi. However, certain X-linked genes, known as escapees, resist this repressive environment and are expressed from the Xa and from the Xi. This results, in most cases, in expression of these genes being higher in females compared with males, at least at the RNA levels, which may have significant implications for physiology, tissue homeostasis, and sex differences (Navarro-Cobos et al, 2020;

Youness et al, 2021). Escapees can be divided into constitutive escapees, which consistently escape XCI in all contexts, and facultative escapees, which exhibit variable escape patterns across different cells, tissues, or individuals.

Knowledge about which genes escape from XCI is available for only a limited number of species and depends on the methods used for their identification (Peeters et al, 2023). The most up-to-date analyses combined RNA-seq and DNA methylation analyses to probe XCI escapees across 12 placental mammalian species (Balaton et al, 2021). This analysis revealed that mice stand as outliers, with only 5% of X-linked genes escaping XCI, while for most species this fraction reaches 10–20%. In marsupials, different proportions of escapees were found in different species: 14% in opossum (Wang et al, 2014) versus 32% in tammar wallaby (Rodríguez-Delgado et al, 2014), but whether this reflects species variation or differences in other parameters such as tissues studied or numbers of informative SNP remains to be evaluated.

The mechanisms underlying this remarkable diversity in the XCI status of X-linked genes remain unclear. Variations in *Xist/RSX* localization, along with differences in chromatin status, may influence which genes are resistant to some extent to silencing. One example of this is the unusual, dispersed localization of *Xist/XIST* in non-activated lymphocytes, which leads to the loss of repressive histone modifications and increased escape of several immune-related X-linked genes from XCI (Wang et al, 2016). This situation is resolved upon lymphocyte activation, as *Xist/XIST* relocates to the Xi through its RBPs, and repressive histone modifications are restored (Syrett et al, 2017; Wang et al, 2016). Furthermore, *Xist* RNA levels can also influence escaping profiles. *Xist* overexpression in mouse neural progenitor cells and embryos results in silencing of escapees, which is accompanied by increased promoter methylation and the disappearance of TAD-like domains that are typically maintained around escapees on the Xi (Hauth et al, 2024). Given these findings, an intriguing question is whether *RSX* RNA levels or distribution could similarly influence the profiles of escapees.

## Diversity in strategies and timing of X chromosome dosage compensation between therian species

### Parental preference versus random XCI

XCI in mammals exists in two forms: imprinted, where the paternal X is preferentially silenced, and random, where either the maternal or paternal X can be inactivated. These two forms of XCI are species-specific but can also vary across cell types within the same species (Fig. 2). For instance, during mouse preimplantation development, imprinted XCI (iXCI) is established up to the blastocyst stage (Harper et al, 1982; Huynh and Lee, 2003; Kay et al, 1993). After this stage, iXCI is maintained only in the trophectoderm and the primitive endoderm, forming the extra-embryonic annexes, while it is reversed in the cells of the inner cell mass, where random XCI (rXCI) subsequently occurs (Mak et al, 2004; Okamoto et al, 2004). Both forms of XCI depend on *Xist* expression from the future Xi (Marahrens et al, 1997; Yang et al, 2016; Wutz and Jaenisch, 2000; Penny et al, 1996; Borensztein et al, 2017). iXCI is also observed in the extraembryonic tissues and across all fetal and adult tissues of marsupials (Mahadevaiah et al, 2020; Sharman, 1971; Wang et al, 2014).

The existence of an imprinted form of XCI in eutherian species other than rodents has long been subject to debate. Early studies in

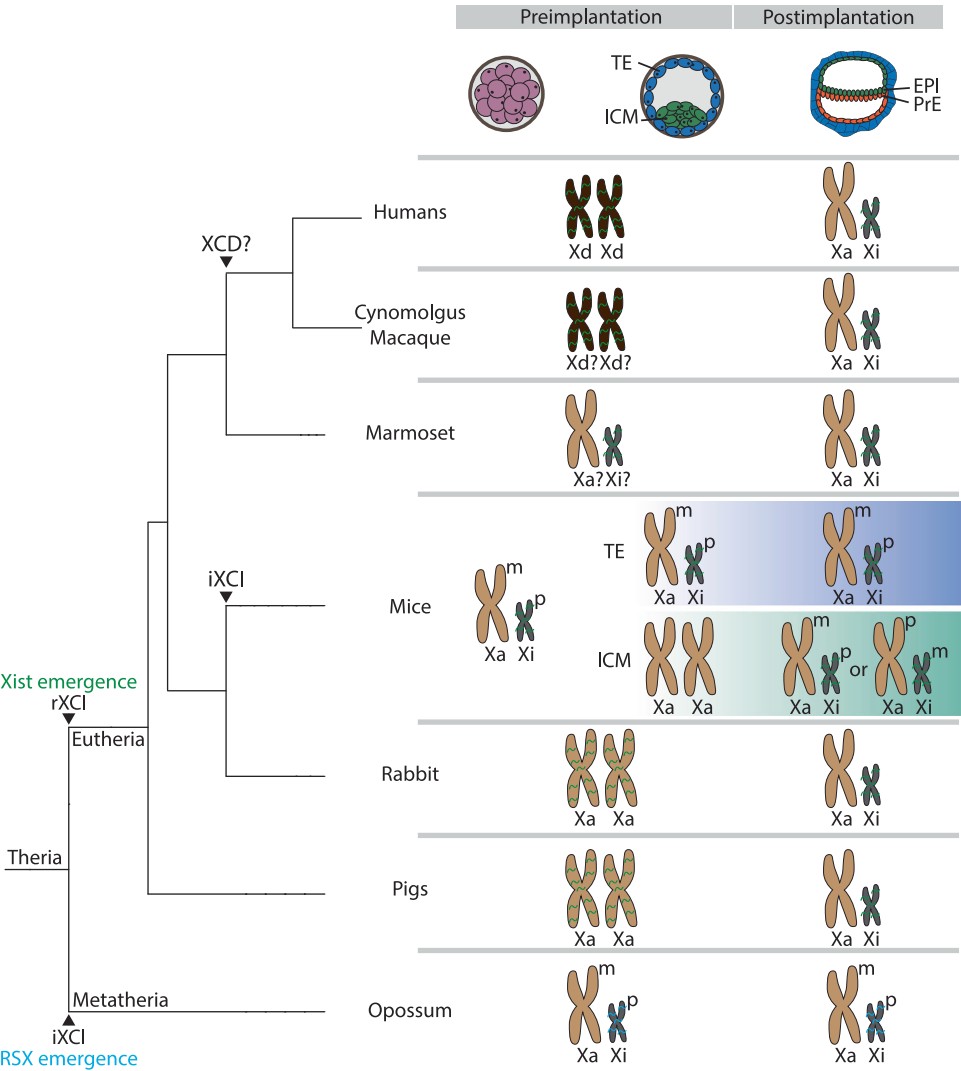

**Figure 2. Diversity in XCI strategies and timing across mammals.**

Left: a phylogenetic tree illustrating the evolution of random and imprinted XCI, XCD, and the emergence of lncRNAs *Xist* and *RSX* in Theria. Right: Schematic illustration of the dynamics of XCD and XCI during pre- and post-implantation development in representative mammalian species. Maternal (m) and paternal (p) X chromosomes are indicated in species with iXCI. *Xist* and *RSX* lncRNAs are colored green and cyan, respectively. TE trophectoderm, ICM inner cell mass, EPI epiblast, PrE primitive endoderm. Active (Xa), dampened (Xd) and inactive (Xi) X chromosomes are represented with different colors and sizes, with Xa in light brown, Xd in dark brown, and Xi in gray. Adapted from Furlan and Galupa, 2022.

humans based on the analysis of a limited number of polymorphic X-linked loci suggested that extraembryonic tissues undergo iXCI, with preferential paternal inactivation (Goto et al, 1997; Harrison, 1989; Harrison and Warburton, 1986; Ropers et al, 1978). ScRNA-seq analysis of the placenta and peri-implantation embryos however revealed the random nature of XCI in extraembryonic and embryonic tissues (Mello et al, 2010; Zhou et al, 2019). This discrepancy with previous studies arose from the organization of placental tissues into large patches of cells, where either the maternal or paternal X chromosome is inactive, unlike other tissues that exhibit mosaic XCI (Mello et al, 2010; Phung et al, 2022). The fact that *XIST* is expressed from both parental alleles throughout human preimplantation development indicates further that *XIST* expression is not imprinted in human early embryos (Okamoto

et al, 2011). Similar findings were made in rabbit and pig embryos, arguing against iXCI, even if the XCI allelic status has not been looked at in rabbit placentas (Okamoto et al, 2011; Zou et al, 2019). rXCI was also demonstrated in mule and horse placenta through RNA-seq and pyrosequencing profiling of numerous X-linked gene expression (Wang et al, 2012). In contrast, it was concluded, based on allelic expression of *XIST* and the *MAOA* X-linked gene, that iXCI, with preferential inactivation of the paternal X chromosome, takes place in bovine placental tissues (Xue et al, 2002). Based on what was found in humans, these results must be interpreted with caution. Further studies are required for apprehending in full the spread of extraembryonic iXCI in placental mammals. However, among placental mammals, iXCI appears to be the exception rather than the rule.

### Timing and dynamics of XCI: early establishment and species-specific variation

XCI is established early during mammalian embryonic development, with timing and kinetics varying between species (Fig. 2). The imprinted form of XCI initiates earlier than rXCI. In mice, iXCI begins around the 2- to 4-cell stage, while in marsupials it initiates at the 8-cell stage, both coinciding with zygotic genome activation (ZGA) (Borensztein et al, 2017; Mahadevaiah et al, 2020). In contrast, rXCI in mice does not commence until approximately day 5.5 of development, when the blastocyst implants (Cheng et al, 2019; Mak et al, 2004; Wang et al, 2017). Both forms of XCI are initiated asynchronously across cells, in a lineage-dependent manner, and unlike iXCI, rXCI is not yet fully completed by E6.5 (Cheng et al, 2019; Borensztein et al, 2017).

Studies utilizing human preimplantation embryos confirmed the lack of XCI at these stages and revealed significant differences in X chromosome features compared with mice. Indeed, *XIST* can be detected following ZGA around the 8-cell stage and coats every X chromosome in female preimplantation embryos, but discrepancies remain as to its expression in male embryos at this stage (Briggs et al, 2015; Daniels et al, 1997; Okamoto et al, 2011; Petropoulos et al, 2016; Ray et al, 1997; Vallot et al, 2017). Notably, *XIST* coating at this stage does not trigger X-linked gene silencing (Okamoto et al, 2011; Petropoulos et al, 2016; Vallot et al, 2017). Similar observations were made in rabbit and cynomolgus monkey preimplantation embryos, altogether revealing an uncoupling between *XIST* expression and XCI (Okamoto et al, 2021, 2011). Intriguingly, even though the presence of *XIST* does not trigger silencing, the active X chromosomes are decorated with repressive histone modifications (Alfeghaly et al, 2024; Okamoto et al, 2021).

The precise timing of XCI onset in humans remains uncertain. ScRNA-seq analysis of human embryos cultured in vitro from day 6 to day 12 suggested that XCI appears to commence around the implantation period (day 7), starting in the trophectoderm and subsequently progressing to the epiblast and primitive endoderm (Zhou et al, 2019). By day 12, XCI is still incomplete. Similar XCI timing and dynamics were observed in cynomolgus monkeys, indicating a longer timeframe for XCI establishment in primates compared with mice (Okamoto et al, 2021). A similar trend was reported in pigs, where single cell analyses of pig embryos, from morula (~E4-5) to spherical embryos (~E10-11), suggested that XCI occurs only in later stages (Ramos-Ibeas et al, 2019). *XIST* expression, as in rabbits and primates, appears to precede XCI onset by several days, as it is already detected in the morula, albeit in females only. Additional cross-species examination in a broader range of mammals is needed to fully understand the reasons for such diversity in XCI timing and dynamics. This variability may be linked to gestation time and to developmental dynamics, such as the timing of ZGA, which occurs much earlier in mice and marsupials compared with other species.

### Is XIST involved in alternative dosage compensation mechanisms?

The absence of XCI during the preimplantation window in various eutherian species raises important questions about alternative mechanisms balancing X-linked gene dosage between males and females before XCI establishment, and the role of *XIST* in early development. Allelic analysis of X-linked genes from scRNA-seq data of human preimplantation embryos revealed that, between E4 and E7, the number of genes with biallelic expression remained constant, while their expression levels were overall reduced to match the levels observed in males (Petropoulos et al, 2016). This suggested that a dosage compensation mechanism, referred to as X chromosome dampening (XCD), might transiently occur in females. There has been a debate as to whether XCD is real or whether it is a computational artifact (Moreira de Mello et al, 2017; Reinius and Sandberg, 2019). One way of addressing this issue is to perturb the system, which, for ethical reasons, is rather to be done in cellular models of human early embryos. Naive human embryonic stem cells (hESCs) display characteristics of preimplantation epiblast cells, including two active X chromosomes, *XIST* expression with X chromosome coating, and the accumulation of repressive histone modifications on the X (Sahakyan et al, 2017), and were proposed to also recapitulate XCD (Sahakyan et al, 2017). Of note, an artifact of this cellular system is that in most naive hESCs, *XIST* is expressed and covers only one of the 2 active X (Sahakyan et al, 2017; Vallot et al, 2017), a property that has been recently exploited to explore the role of *XIST* in XCD. Evidence based on RNA-FISH and allelic RNA-seq analyses indicated that genes on the X chromosome coated by *XIST* are expressed at a lower level compared with the other X. More strikingly, removal of *XIST* or of its protein partner SPEN abolishes this asymmetry and results in two equally expressed X chromosomes (Alfeghaly et al, 2024; Dror et al, 2024). These findings strengthen the conclusion that expression of X chromosomes can be attenuated in models of human early development, identify *XIST* as a key mediator of XCD, and demonstrate that both XCI and XCD rely, at least partially, on the same set of factors.

It is still uncertain whether XCD is specific to hominoids or occurs in other primates and in more distantly related species with late XCI onset. Pereira and colleagues concluded there is a lack of XCD in marmoset (Callithrix jacchus) preimplantation embryos based on the analysis of embryonic scRNA-seq datasets (Cidral et al, 2021). Instead, they reported an increase of monoallelically expressed X-linked genes in female embryos between the morula and late blastocyst stages, indicating early XCI onset. A more nuanced interpretation of cynomolgus preimplantation embryo data was proposed by Okamoto and collaborators, where a dampening-like process was proposed, even if not leading to equal expression levels of X chromosomes in females and males (Okamoto et al, 2021). Finally, dosage compensation in pig embryos appears to be achieved solely by XCI, thus excluding XCD (Ramos-Ibeas et al, 2019). More work is needed to conclude as to the existence, the breadth across mammals and, more importantly, the function of XCD.

## Regulation of XCI: when additional noncoding elements come into play

The *XIST* gene resides in a genomic region known as the X inactivation center (*Xic*), which is both essential and sufficient to initiate X chromosome silencing (Rastan and Robertson, 1985). This region is rich in noncoding elements, especially lncRNAs (e,g, *Tsix*, *Jpx*, *Ftx*, *Linx*), that play crucial roles in regulating *Xist*/*XIST* expression (Fig. 3). While the overall synteny, order, and orientation of genes within the *Xic* are conserved across species like mice and primates, differences exist in its 3D organization and in the contribution of certain noncoding components. In contrast, the marsupial *Xic* remains poorly characterized, although emerging

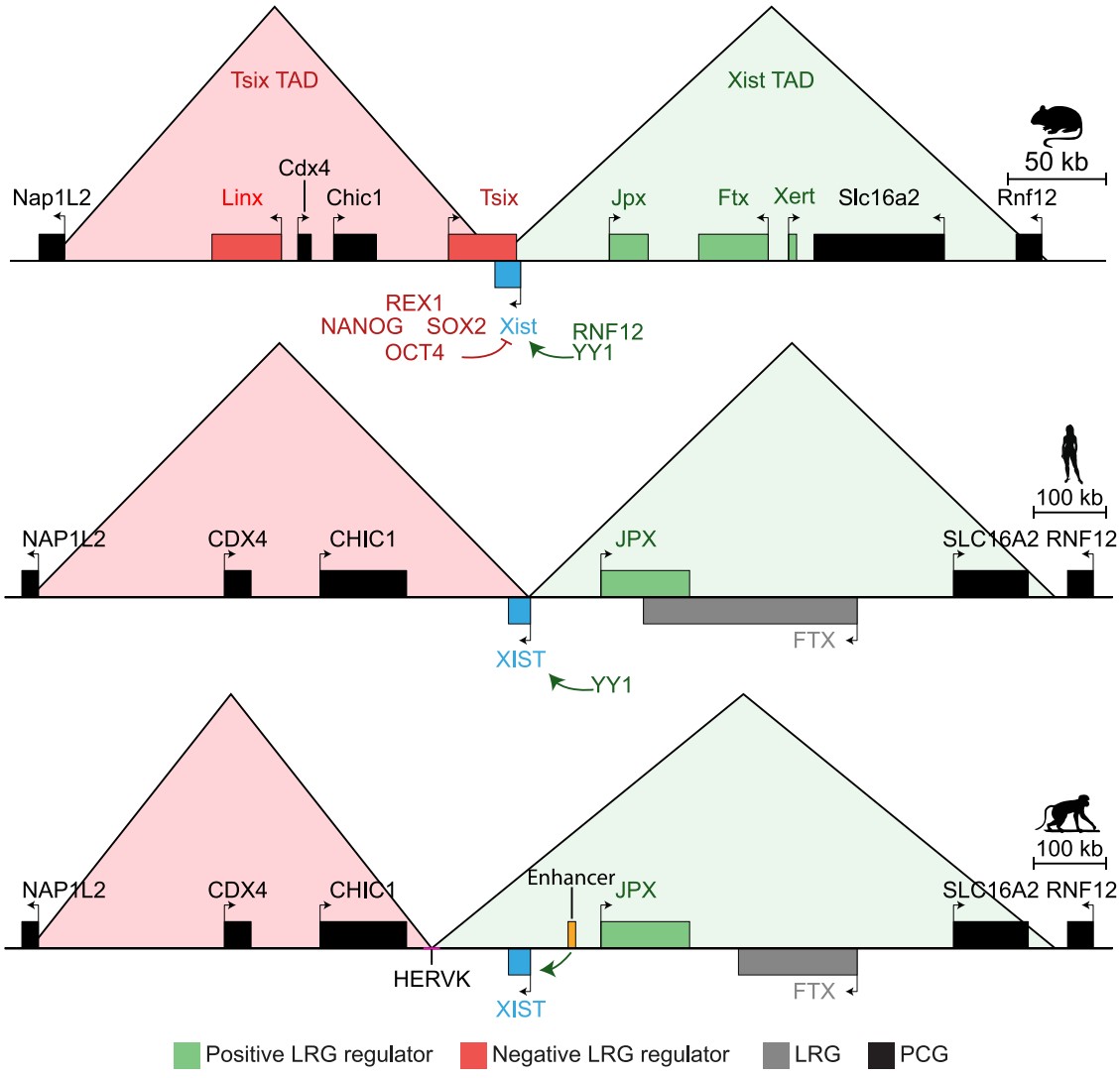

**Figure 3. *Xic* chromatin conformations in mice, humans, and rhesus macaques.**

Schematic representation of the chromatin regulatory landscape of the *Xist/XIST* gene across different species, highlighting the conserved synteny and variations in long noncoding RNA genes (LRGs), protein-coding genes (PCGs), noncoding elements, and TAD organization. Autosomal factors influencing *Xist/XIST* expression, either positively or negatively, are also shown.

evidence points to the presence of antisense regulatory mechanisms reminiscent of those observed in mice (see below).

### Antisense regulation at the heart of imprinted XCI?

*Xist* is transcribed antisense to another lncRNA called *Tsix*, which was first identified as a negative regulator of *Xist* expression in mice (Lee et al, 1999). At the onset of rXCI, *Tsix* influences which X chromosome remains active by preventing *Xist* upregulation in *cis* through various mechanisms that have been extensively discussed (Furlan and Rougeulle, 2016; Sado, 2017). In iXCI, *Tsix* also plays a role by inhibiting the inactivation of the maternal X chromosome during the differentiation of trophectoderm cells (Maclary et al, 2014). However, *Tsix* is not the primary factor determining the imprint that renders the maternal X chromosome resistant to silencing. Instead, the biased expression of *Xist* from the paternal X chromosome results from a noncanonical form of parental imprinting, whereby the maternal allele

is repressed by H3K27me3, a mark inherited from the oocyte and maintained throughout preimplantation development (Inoue et al, 2017). This mark is lost in the epiblast cells, enabling the transition to random monoallelic expression of *Xist* (Inoue et al, 2017).

Only a truncated version of *TSIX* was reported in humans, but its function, if any, has not been determined (Migeon et al, 2001). In human fetal cells, truncated *TSIX* was found to be transcribed from the Xi and co-expressed with *XIST*, arguing against an antagonistic function, and no expression of *TSIX* has been detected so far in early embryonic stages when XCI initiates (Migeon et al, 2002). Furthermore, *TSIX* has not been found in the genomes of other mammals such as dogs, cats, cows, sheep, pigs, guinea pigs, ferrets, alpacas, and sea otters (Aksit et al, 2024). This strongly suggests that *Tsix* function in XCI is specific to rodents.

The mechanisms regulating *RSX* expression and iXCI in marsupials remain to be fully elucidated. Differential DNA

methylation has been observed on the *RSX* promoter, with methylation occurring exclusively on the maternal active X. However, it remains to be determined whether this methylation influences *RSX* expression (Wang et al, 2014). Interestingly, inspection of scRNA-seq datasets identified an RNA that overlaps and is transcribed antisense to *RSX*, named *XSR* (Mahadevaiah et al, 2020). *XSR* is exclusively expressed from the maternal X chromosome as opposed to *RSX*, and only in embryos. This is reminiscent of *Tsix* in mice which makes *XSR* an interesting candidate for controlling iXCI in marsupials. Future experiments that perturb *XSR* expression will be crucial in testing this hypothesis. The presence of antisense noncoding transcripts in both marsupials and mice raises the question of whether there is a connection between such paired organization and the occurrence of iXCI in rodents and marsupials. It remains to be determined whether other noncoding RNA genes are present within the marsupial *Xic*-like region and contribute to XCI regulation in these species.

### Variation in XIST regulation among placental mammals

*Tsix*'s role in regulating *Xist* expression is just one aspect of a much more complex network. The expression of *Xist*/*XIST* is orchestrated by a variety of positive and negative regulators, whose mechanisms of action can vary between eutherian species. For instance, the promoter of a lncRNA named *Linx* (LinxP) functions as a long-range silencer of *Xist*, operating independently of *Linx* transcription and without impacting *Tsix* expression (Galupa et al, 2020). While LinxP is conserved in sequence and synteny across placental mammals, there is no evidence of transcription at this locus in humans, and its potential silencing role in other species remains unclear.

Among *Xist*'s key positive regulators in mice are the lncRNA genes *Jpx* and *Ftx*. *Jpx* is conserved in synteny rather than sequence (Karner et al, 2020), while *Ftx* exhibits synteny, intron-exon organization and sequence conservation, the latter being biased towards the *Ftx* 5' end (Chureau et al, 2011). Despite such substantial conservation, *Ftx* was shown to regulate *Xist* expression in mice, but not in humans (Furlan et al, 2018; Rosspopoff et al, 2023). In contrast, functional conservation of *JPX* between humans and mice has been proposed since the phenotype of *Jpx* knockout mice could be rescued using the human ortholog (Karner et al, 2020). The role of *JPX* in XCI has been confirmed in both species through a range of functional approaches, which, however, revealed distinct modes of action (Rosspopoff et al, 2023; Tian et al, 2010). In mice, the *Jpx* RNA molecule regulates *Xist* expression in *trans*, although the exact mechanisms remain controversial, whereas in humans, it is the transcription of the *JPX* locus in *cis*, rather than the RNA itself, that is promoting *XIST* expression (Rosspopoff et al, 2023; Sun et al, 2013). A recent study in rhesus macaque ESCs revealed that *JPX*'s regulatory role on *XIST* is reduced compared to humans and involves cooperation with a macaque-specific enhancer (Cazottes et al, 2023). Other enhancers within the *Xic* were recently discovered in mice, through a pooled CRISPRi screen, as crucial for *Xist* upregulation in *cis* at the onset of XCI (Gjaltema et al, 2022). Specifically, a cluster of enhancers within the intron of the lncRNA *Xert* forms a regulatory hub together with *Ftx* and *Xist* promoters (Gjaltema et al, 2022). These elements exhibit characteristics of active enhancers in hESCs, suggesting functional conservation.

Beyond X-linked regulators, autosomal factors also play a crucial role in regulating XCI (Schwämmle and Schulz, 2023). In

mice, pluripotency factors, such as REX1, NANOG, SOX2, and OCT4, were proposed to negatively regulate *Xist* expression (Navarro et al, 2010, 2008) but whether such a control exists beyond rodents is currently unknown. The stability of REX1 is controlled by the E3 ubiquitin ligase RNF12, encoded by a gene located in the *Xic* (Gontan et al, 2018). Conversely, the transcription factor YY1 positively regulates *Xist*/*XIST* expression in both humans and mice, and likely in other eutherians (Makhlouf et al, 2014).

### Variation in Xic 3D organization among placental mammals

The noncoding regulators of *Xist* discussed above are organized in a highly structured manner, being physically separated into two distinct TADs (Fig. 3) (Nora et al, 2012). In mice, all *Xist* activators are located within one TAD which also houses the *Xist* promoter, while the adjacent TAD contains the negative regulators (Galupa and Heard, 2018). This precise organization has been shown to contribute to correct regulation of *Xist* levels (van Bemmel et al, 2019). Comparative analyses from mice, humans, chimpanzees, and rhesus macaques have revealed divergence in the *Xic* TAD organization across species (Cazottes et al, 2023; Rosspopoff et al, 2023). In humans and chimpanzees, the proximal and distal boundaries of the *XIST*-associated TAD are located near the *XIST* 5' promoter region and downstream of the *SLC16A2* gene, respectively. In mice, the *Xist*-associated TAD includes the entire *Xist* gene on one end, and the *Rfn12*/*Rlim* gene on the other (Rosspopoff et al, 2023). In rhesus macaques, the proximal boundary of the *XIST*-associated TAD is shifted downstream of the CHIC1 gene, triggered by a HERVK insertion containing four CTCF binding sites that promote barrier activities (Cazottes et al, 2023). The deletion of this HERVK in rhesus macaques realigns the 3D organization of the *Xic* with that of humans, without significantly affecting *XIST* expression. Proper control of *Xist* expression in eutherians thus involves a complex and dynamic interplay of noncoding elements that diverged substantially in function and spatial organization even over short timescales.

### Existence of additional XCI regulators in humans?

*XIST* expression patterns in early human embryos raise questions as to the existence of *XIST* antagonists, which would prevent it from triggering XCI at these stages. The hominoid-specific *XACT* lncRNA has been proposed to play such a role (Vallot et al, 2017). *XACT* is expressed in early human developmental stages from the long arm of the X chromosome, approximately 50 Mb from *XIST* (Vallot et al, 2013). What makes *XACT* unique is its ability to accumulate in *cis* and form a cloud on the Xa, mirroring *XIST*'s accumulation on the Xi (Vallot et al, 2013). During human preimplantation development and in naive hESCs, *XIST* and *XACT* RNAs co-accumulate on active Xs, although they occupy distinct territories (Vallot et al, 2017). The introduction of an *XACT* transgene on one X chromosome in mouse ESCs skewed the XCI choice towards the allele not carrying the transgene upon differentiation, which led to the antagonist hypothesis, in which *XACT* would be protecting the X chromosome from silencing by influencing *XIST* expression, localization, or activity (Vallot et al, 2017). This hypothesis has however been discarded by recent findings showing that the loss of *XACT* in naive hESCs did not trigger XCI and had no effect on *XIST* expression nor distribution (Alfeghaly et al, 2024). Deletion of *XACT* in primed hESCs did not

**Box 1   In need of answers**

1. What factors drive the diverse patterns of XCI escape across mammals, and how might variations in escapee status impact sex-specific fitness and evolutionary trajectories?
2. What prevents *XIST* from silencing both X chromosomes in human preimplantation embryos, and what molecular events trigger the transition in its activity, from dampening to full silencing?
3. Is X chromosome dampening essential for proper embryo development, and what are the consequences of its disruption?
4. How critical is XCI in species beyond the well-studied mouse model, and what roles does it play in other mammals, including primates and marsupials?

affect X-linked gene dosage either (Motosugi et al, 2021). It did impact, however, the neural differentiation pathway through mechanisms that remain to be determined. Whether *XACT* intervenes in other aspects of XCI or in independent processes in humans requires further investigation.

The involvement of other X-linked or autosomal *trans*-acting regulators of *XIST* activity in humans has yet to be resolved. Future studies based on screening approaches will likely be instrumental in identifying factors involved in X chromosome silencing.

## Discussion and concluding remarks

Our comprehension of XCI mechanisms, a quest initiated over 60 years ago with the seminal discovery by British geneticist Mary Lyon (Lyon, 1961), has reached remarkable resolution in recent years and uncovered surprising variability in how this process is established, not only between marsupials and placental mammals, but also among eutherians themselves. This variability is particularly striking when considering the kinetics of XCI, the main regulatory players, such as *Xist* in eutherians and *RSX* in marsupials, and the presence of iXCI exclusively in rodents and marsupials.

Understanding why different forms of XCI, imprinted and random, emerged during mammalian evolution remains a challenge. One hypothesis posits that rXCI promotes cellular mosaicism, which may confer a selective advantage by increasing resilience to deleterious X-linked mutations (Franco and Ballabio, 2006; Ng et al, 2007). This potential benefit could have driven its widespread adoption in placental mammals. In contrast, marsupials and the extraembryonic tissues of rodents may compensate for the absence of rXCI through a higher proportion of escapee genes, allowing the expression of both X-linked alleles in females and thus providing an alternative safeguard against harmful mutations.

Another non mutually exclusive possibility is that iXCI originally evolved to prevent the accidental silencing of both X chromosomes in the early embryo by tightly regulating the expression of *Xist* and *RSX* from the onset of ZGA. This raises the intriguing idea that in species like humans, which do not have iXCI, alternative mechanisms like X chromosome dampening may instead have evolved to compensate for biallelic *XIST* expression, by limiting gene dosage without fully shutting down both X

chromosomes. It's also worth considering that iXCI may simply result from species-specific epigenetic mechanisms. As mentioned above, iXCI in mice relies on noncanonical imprinting pathways involving maternal chromatin marks. So far, there's no evidence for a similar mechanism in humans or other primates (Albert and Greenberg, 2023), suggesting that this form of imprinting may be unique to rodents.

In conclusion, mice have long been considered the leading model for studying XCI. However, the differences across species, as highlighted in this review, urge us to reconsider the universality of our current understanding. Incorporating additional model organisms into XCI studies is crucial to answering key questions (see Box 1) and advancing our knowledge. The coming years promise to provide groundbreaking insights into this complex process.

## Peer review information

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

## Acknowledgements

The authors would like to apologize to all those authors whose findings we could not cite due to space limitations. The authors thank Celine Morey for critically reading the manuscript. Work in the Rougeulle lab is funded by the Agence Nationale de la Recherche (ANR-14-CE10-0017) and the European Research Council (ERC) under the European Union's Horizon 2020 research and innovation program (ERC AdG 101020423).

## Author contributions

**Charbel Alfeghaly**: Conceptualization; Writing—original draft; Writing—review and editing. **Claire Rougeulle**: Conceptualization; Resources; Supervision; Writing—review and editing.

## Disclosure and competing interests statement

The authors declare no competing interests.

