## [Peer Review File · EMBO Reports]

X chromosome inactivation in mammals: general principles and species-specific considerations

Charbel Alfeghaly and Claire Rougeulle

Corresponding author(s): Claire Rougeulle (claire.rougeulle@curie.fr)

Review Timeline:

Submission Date:	18th Feb 25
Editorial Decision:	20th Mar 25
Revision Received:	9th May 25
Accepted:	3rd Jun 25

Editor: *Esther Schnapp*

Transaction Report:

Dear Claire,

Thank you for the submission of your Review to EMBO reports. We have received the full set of comments now, which are pasted below.

As you will see, all referees acknowledge that the review is very timely and interesting and of high quality, which is great. I would like to ask you now to address the referee comments as you see fit. When you submit the final version of the review, I will go through the text carefully and edit it as necessary.

With your final review, please upload the figures separately. Only the figure legends need to be in the ms file, after the references. If you like, the figures can be redrawn by our graphics designer, which will take 10 working days after you submit the final review. We can also use the figures directly as you provide them, given their good quality. Please let me know what you prefer.

If you use BioRender or similar to create the figures, this needs to be acknowledged in the figure legends or the Acknowledgements.

On the title page, the corresponding author needs to be clearly marked and your email address needs to be added.

Please provide up to 5 keywords with your review.

The conflict of interest statement is good, but it needs the "Disclosure and Competing Interests Statement" heading.

The REFERENCE FORMAT needs to be corrected: et al needs to be used after 10 author names; DOIs should only be used for preprints and datasets that have not been published yet. Please use the EMBO reports reference style.

We ask our authors to include in their reviews a 'Box' called "In need of answers" that briefly outlines the major questions that are still open in a given field. These questions can be accompanied by suggested experimentation that addresses the questions.

As for timing, would it be possible for you to submit the revised version by April 21? If you anticipate a problem meeting this date, then please just let me know.

Thank you again for contributing this nice piece to EMBO reports !

With best wishes,
Esther

Referee #1:

This is a timely, balanced, very relevant and very well-written review that discusses the current knowledge in the field of X-inactivation in light of the diversity observed across mammalian species. The authors did a great job in integrating the findings from different species, which will certainly be useful not only to many in the field but also to other researchers interested in learning from about X-inactivation.

I have just a few suggestions that I think could help polish some points:

1. The second paragraph of the introduction, on the evolution of the sex chromosomes, is written in a rather categorical manner. Given that we can only infer what has happened, I would suggest making it clear that the way we think it happened is "just" an hypothesis.
2. In the third paragraph the authors refer to mouse but XIST is written with the human nomenclature.
3. This sentence could be revisited: "XIST lncRNA is the master regulator of XCI in eutherians." I think this is only unequivocally shown for the mouse? At most, for mouse and human? (though strictly speaking, in humans XIST has only been genetically implicated with XCD and not XCI, if I am not mistaken)
4. "Repeat A is indispensable for gene silencing, and it is the only repeat with nucleotides under selective constraint in mammals

based on PhyloP scores, highlighting its crucial role in XCI." I think this information benefit from a reference; if it was analysis done by the authors, it could be shown in a figure (which would have the potential to be used by many in the field).

5. "It is worth noting that SPEN was not identified in the Rxs interactome" - maybe a small note just to reassure the reader that there is an orthologue of SPEN in marsupials.

Congratulations to the authors on this very interesting review!

Referee #2:

The review "X chromosome inactivation in mammals: general principles and species-specific considerations", submitted to EMBO Reports, by Alfeghaly and Rougeulle, is well-written, clear, and up-to-date. The authors discuss X-chromosome dosage in mammals, with a specific focus on the evolutionary aspects of X chromosome inactivation, the conserved mechanisms and species-specific variations across mammalian species, highlighting knowledge in humans when available.

I have only minor comments which I believe would benefit the readers.

1) For clarity, please name Xist and Rxs lncRNAs when discussing "distinct lncRNAs of independent evolutionary origin" (page 2).

2) Page 7: What does Xi-elect mean? "At the onset of XCI, the eutherian and marsupial Xi-elect becomes depleted in histone modifications associated with active transcription and enriched in repressive histone modifications."

3) Missing references:

- Page 10: "Both forms of XCI depend on Xist expression from the future Xi"

- Page 11: "XCI is established early during mammalian embryonic development, with timing and kinetics varying between species (Figure 2). The imprinted form of XCI initiates earlier than rXCI. In both mice and marsupials, iXCI begins as early as the 2- to 4-cell stage, coinciding with zygotic genome activation (ZGA). In contrast, rXCI in mice does not commence until approximately day 5 of development, when the blastocyst implants."

- Page 11: Problem with references? "Indeed, XIST can be detected following ZGA around the 8-cell stage (79-81) and coats every X chromosome in female preimplantation embryos"

4) The authors suggest that random XCI in cynomolgus monkeys takes longer than in mice. However, speed of random XCI in mice seems heterogeneous, cell-type specific, and could take longer than previously expected, at least for some cells (not yet silence at 6.5dpc, Q. Deng lab, 10.1016/j.celrep.2019.02.031); page 11. This also needs to be considered in the context of each species' gestational timing.

5) Figure 2: blastomere from the morula stage should be rounder for mammalian morula. As TE and ICM are mentioned, Epi and PrE should also be stated in the late blastocyst.

6) Page 14: The authors introduce the H3K27me3 imprint of Xist in mice. It would help to clarify that this imprint is later lost during development to allow random monoallelic expression of Xist in peri-implantation embryos.

7) Page 15: "This strongly suggests that Tsix is specific to rodents." Because the authors report the existence of a truncated TSIX in humans, I would suggest modifying this conclusion to "This strongly suggests that Tsix function in XCI is specific to rodents"

Referee #3:

In their review, Alfeghaly and Rougeulle summarize recent literature advancements in XCI research from a mammalian species-centric view. They address a wide range of topics including different lncRNAs, protein interactions, epigenetic modifications, DNA architecture, etc. Their review is timely and will likely be a good resource in the field for general overview.

I had only one major comment. Given that this is a cross-species comparison, it would be useful for the authors to speculate more on why XCI mechanisms vary so much between species. For instance, why is the timing of XCI so different between species? Why do some organisms silence both Xs and others have strictly monoallelic silencing? Why do some mammals have imprinted and others random XCI? Some speculation would elevate the review above a very good summary of published findings.

Minor comments:

1. Row 26: Therian sex chromosome originated between 160-166 MY ago, and not 180 MY ago, as stated.

2. Row 42: Xist transcript should be with lower case letters (mouse). Please ensure correct formatting throughout the text as (multiple examples, such as LNX for mouse protein (row 86)).

3. Row 56: The term "DNA elements" is ambiguous. Do you mean promoters/enhancers? What do you mean "dependence"? XCI can also occur when controlled from Dox-induced promoter, or translocated onto another chromosome. Please rephrase.

4. Row 59: RSX should be upper case, italicized throughout the text (opossum transcript format).

5. Row 87: "factors" is a broad term; did you mean to say lncRNA? Some proteins (also factors) can be shared, some can be unique. Please rephrase this.

6. Row 97-99: In this paragraph it'll be advised to mention that there is no sequence similarity between RSX and Xist.
7. Row 143: McIntyre et al., citation is not about Xist interactome, as the preceding sentence would suggest.
8. Row 179: The study on which these findings are based, McIntyre et al, presents with methodological issues including scarcity of validation and weak evidence to substantiate claims. Perhaps more scepticism is warranted in raising specific proteins (e.g CKAP4), and future research can revisit and deepen understanding of this question.
9. Row 203-204: The Xi-elect becomes depleted in histone modifications associated with active transcriptions at XCI onset in eutherians, but this to my knowledge hasn't been studied in marsupials.
10. Please consider citing Rens et al., 2010, PNAS article as an important study that profiled different acetylations and methylations in opossum X.
11. Row 261: Regarding the "no overlap" in XCI escapees, consider mentioning that this is in accordance with the major escapees being Y-linked gametologs with somatic expression. Since the murine and marsupial Y chromosomes diverged significantly (Bellot et al., Nature, 2014), it makes sense that the corresponding escapees will too.
12. Row 276: It's unlikely that the "absence of DNA methylation on the Xi contributes to a different pattern of XCI escape" because it is throughout the chromosome. Please see comment 11 about escapees.
13. Row 293: Regarding iXCI in opossums being the only form of XCI in all tissues - consider mentioning that marsupial pluripotency is decoupled from XaXa status, which complements the paragraph's theme of iXCI reversal in the mouse pluripotent inner cell mass.

At the end of this / subsequent paragraph would be a good place (among others) to discuss theories and trends (see major comment above).

14. Rows 320-321: ZGA and iXCI starts around 8 cell stage in opossum, and not as mentioned.
15. Row 325: References 79-81 appear in different format.
16. Row 335: The statement that Xist may have "functions beyond XCI" due to the findings above, doesn't make sense. The accumulation of Xist/proteins/histone modifications in absence of transcriptional silencing, can be explained by the absence of a critical and unknown key component that makes all elements "work together", or by specially restricted accumulations of these elements (e.g intergenic area only) that cannot be further resolved by microscopy only. See again, major comment, hypothesis, thought.
17. Row 392: Define the different colours that Xa, Xd and Xi are represented with in this legend.
18. Row 403-404: I'm not sure I agree that a marsupial Xic-like region has not been clearly defined. The Xic is an X-inactivation centre; a site on the future inactive X from which the instructive silencing RNA is produced. Such a site has been demonstrated based on Rsx transgenic studies in mice. I wonder here whether the authors mean that a more complex Xic housing multiple ncRNAs akin to that in eutherians has not yet been defined in marsupials?
19. Regarding the "mechanisms controlling RSX expression" mentioned in row 404, also applying for the statement in row 424: Please note the differential RSX promoter DNA methylation, which is maternally present and paternally absent, from Wang et al., 2014, Genome research, (which is cited in this review) as a likely source of RSX regulation. Also note that the Waters lab recently claimed that imprinted XCI is regulated by inheritance of a hypomethylated inactive X doi: 10.1073/pnas.2412185121
19. 20. Row 453: Please rephrase "which however revealed", this is not clear.

Editor comments:

If you use BioRender or similar to create the figures, this needs to be acknowledged in the figure legends or the Acknowledgements.

NA

On the title page, the corresponding author needs to be clearly marked and your email address needs to be added.

Done.

Please provide up to 5 keywords with your review.

Five key words have been added to the title page.

The conflict of interest statement is good, but it needs the "Disclosure and Competing Interests Statement" heading.

Done.

The REFERENCE FORMAT needs to be corrected: et al needs to be used after 10 author names; DOIs should only be used for preprints and datasets that have not been published yet. Please use the EMBO reports reference style.

Done.

We ask our authors to include in their reviews a 'Box' called "In need of answers" that briefly outlines the major questions that are still open in a given field. These questions can be accompanied by suggested experimentation that addresses the questions.

We added "In need of answers" box after the Discussion and conclusion section.

Referee #1 comments:

This is a timely, balanced, very relevant and very well-written review that discusses the current knowledge in the field of X-inactivation in light of the diversity observed across mammalian species. The authors did a great job in integrating the findings from different species, which will certainly be useful not only to many in the field but also to other researchers interested in learning from about X-inactivation. I have just a few suggestions that I think could help polish some points.

We appreciate and thank the Reviewer's positive feedback on the manuscript and their time spent reviewing it. Below, we address their comments.

The second paragraph of the introduction, on the evolution of the sex chromosomes, is written in a rather categorical manner. Given that we can only infer what has happened, I would suggest making it clear that the way we think it happened is "just" an hypothesis.

We have modified the sentence on line 6 as such: "It is widely hypothesized that the differentiation of sex chromosomes began with a mutation on the proto-Y, which led to the emergence of a male sex-determining gene known as SRY (Graves, 2006)."

In the third paragraph the authors refer to mouse but XIST is written with the human nomenclature. This is now corrected.

This sentence could be revisited: "XIST lncRNA is the master regulator of XCI in eutherians." I think this is only unequivocally shown for the mouse? At most, for mouse and human? (though strictly speaking, in humans XIST has only been genetically implicated with XCD and not XCI, if I am not mistaken).

We agree with the Reviewer that XIST function in triggering XCI has been mainly demonstrated in mice, and to a lesser extent in humans. We have thus modified the sentence on line 39 accordingly: "The central role of *Xist* lncRNA in triggering XCI has been extensively demonstrated in mice, and to a lesser degree in humans, but is believed to be conserved in all placental mammals".

"Repeat A is indispensable for gene silencing, and it is the only repeat with nucleotides under selective constraint in mammals based on PhyloP scores, highlighting its crucial role in XCI." I think this information benefit from a reference; if it was analysis done by the authors, it could be shown in a figure (which would have the potential to be used by many in the field).

We have modified Figure 1 and included PhyloP scores.

"It is worth noting that SPEN was not identified in the R_{sx} interactome" - maybe a small note just to reassure the reader that there is an orthologue of SPEN in marsupials.

We have modified the sentence on line 159 accordingly: "It is worth noting that SPEN was not identified in the reported RSX interactome, even though a SPEN orthologue exists in marsupials."

Referee #2 comments:

The review "X chromosome inactivation in mammals: general principles and species-specific considerations", submitted to EMBO Reports, by Alfeghaly and Rougeulle, is well-written, clear, and up-to-date. The authors discuss X-chromosome dosage in mammals, with a specific focus on the evolutionary aspects of X chromosome inactivation, the conserved mechanisms and species-specific variations across mammalian species, highlighting knowledge in humans when available. I have only minor comments which I believe would benefit the readers.

We thank the Reviewer for their positive feedback and time spent reviewing the manuscript. We address their comments below.

For clarity, please name *Xist* and *Rsx* lncRNAs when discussing "distinct lncRNAs of independent evolutionary origin" (page 2).

We have modified the text on line 27 accordingly: "The most striking example comes from the comparison between placental mammals and marsupials, where XCI is triggered by distinct lncRNAs, *Xist* and *RSX* respectively, which have independent evolutionary origins."

Page 7: What does Xi-elect mean? "At the onset of XCI, the eutherian and marsupial Xi-elect becomes depleted in histone modifications associated with active transcription and enriched in repressive histone modifications."

"Xi-elect" refers to the X chromosome that has been selected for inactivation. We have modified the sentence on line 174 based on the comment of Reviewer 3 as such: "The Xi is generally depleted in histone modifications associated with active transcription and enriched in repressive histone modifications in both eutherians and marsupials (Chaumeil et al., 2011; Rens et al., 2010)".

Missing references:

- Page 10: "Both forms of XCI depend on *Xist* expression from the future Xi"

- Page 11: "XCI is established early during mammalian embryonic development, with timing and kinetics varying between species (Figure 2). The imprinted form of XCI initiates earlier than rXCI. In both mice and marsupials, iXCI begins as early as the 2- to 4-cell stage, coinciding with zygotic genome activation (ZGA). In contrast, rXCI in mice does not commence until approximately day 5 of development, when the blastocyst implants."

- Page 11: Problem with references? "Indeed, *XIST* can be detected following ZGA around the 8-cell stage (79-81) and coats every X chromosome in female preimplantation embryos"

This was corrected.

The authors suggest that random XCI in cynomolgus monkeys takes longer than in mice. However, speed of random XCI in mice seems heterogeneous, cell-type specific, and could take longer than previously expected, at least for some cells (not yet silence at 6.5dpc, Q. Deng lab, 10.1016/j.celrep.2019.02.031); page 11. This also needs to be considered in the context of each species' gestational timing.

We have slightly modified the text line 281: "In contrast, rXCI in mice does not commence until approximately day 5.5 of development, when the blastocyst implants (Cheng et al, 2019; Mak et al, 2004; Wang et al, 2017). Both forms of XCI are asynchronous, depending on the lineage, and unlike iXCI, rXCI is not yet fully completed by E6.5 (Cheng et al, 2019; Borensztein et al, 2017)".

We have also added sentences line 305 to include the Reviewer's comment regarding gestation time: "Additional cross-species examination in a broader range of mammals is needed to fully understand the reasons for such diversity in XCI timing and dynamics. This variability may be linked to gestation time and to developmental dynamics, such as the timing of ZGA, which occurs much earlier in mice and marsupials compared to other species."

Figure 2: blastomere from the morula stage should be rounder for mammalian morula. As TE and ICM are mentioned, Epi and PrE should also be stated in the late blastocyst.

The figure will be modified accordingly.

Page 14: The authors introduce the H3K27me3 imprint of Xist in mice. It would help to clarify that this imprint is later lost during development to allow random monoallelic expression of Xist in peri-implantation embryos.

We have added this in the revised version line 365 as such: "Instead, the biased expression of Xist on the paternal X chromosome results from a non-canonical form of parental imprinting, whereby the maternal allele is repressed by H3K27me3, a mark inherited from the oocyte and maintained throughout preimplantation development (Inoue et al., 2017). This mark is lost in the epiblast cells, enabling the transition to random monoallelic expression of Xist (Inoue et al., 2017)."

Page 15: "This strongly suggests that Tsix is specific to rodents." Because the authors report the existence of a truncated TSIX in humans, I would suggest modifying this conclusion to "This strongly suggests that Tsix function in XCI is specific to rodents"

This was modified as suggested.

Referee #3 comments:

In their review, Alfeghaly and Rougeulle summarize recent literature advancements in XCI research from a mammalian species-centric view. They address a wide range of topics including different lncRNAs, protein interactions, epigenetic modifications, DNA architecture, etc. Their review is timely and will likely be a good resource in the field for general overview.

We thank the Reviewer for their contribution and time spent reviewing the manuscript. We address their comments below.

I had only one major comment. Given that this is a cross-species comparison, it would be useful for the authors to speculate more on why XCI mechanisms vary so much between species. For instance, why is the timing of XCI so different between species? Why do some organisms silence both Xs and others have strictly monoallelic silencing? Why do some mammals have imprinted and others random XCI? Some speculation would elevate the review above a very good summary of published findings.

We added a discussion section at the end of the review to speculate on this aspect.

Minor comments:

Row 26: Therian sex chromosome originated between 160-166 MY ago, and not 180 MY ago, as stated.

We have modified the text on line 4 accordingly.

Row 42: Xist transcript should be with lower case letters (mouse). Please ensure correct formatting throughout the text as (multiple examples, such as LNX for mouse protein (row 86)).

Done.

Row 56: The term "DNA elements" is ambiguous. Do you mean promoters/enhancers? What do you mean "dependence"? XCI can also occur when controlled from Dox-induced promoter, or translocated onto another chromosome. Please rephrase.

DNA elements refer to repeats within Xist and RSX. We modified the text accordingly on line 34: "A shared feature of XCI across species is its reliance on non-coding RNAs containing repeat elements, which may act as functional domains."

Row 59: RSX should be upper case, italicized throughout the text (opossum transcript format).

This was modified accordingly.

Row 87: "factors" is a broad term; did you mean to say lncRNA? Some proteins (also factors) can be shared, some can be unique. Please rephrase this.

We meant to keep it broad as it could be lncRNA or proteins, so we suggest keeping the generic term "factor".

Row 97-99: In this paragraph it'll be advised to mention that there is no sequence similarity between RSX and Xist.

We have modified the text on line 75 accordingly: "Despite the distinct evolutionary origins and lack of sequence similarity between Xist and RSX, some features are shared."

Row 143: McIntyre et al., citation is not about Xist interactome, as the preceding sentence would suggest.

We cited this paper since they also performed enrichment analysis.

Row 179: The study on which these findings are based, McIntyre et al, presents with methodological issues including scarcity of validation and weak evidence to substantiate claims. Perhaps more scepticism is warranted in raising specific proteins (e.g CKAP4), and future research can revisit and deepen understanding of this question.

We have written the sentence in a more careful way : line 144 "This disparity could again be due to differences in the cell types and XCI stages studied across species, or to experimental artifacts, which would require additional validation " and line 150 "Depletion of CKAP4 led to an increased proportion of cells with biallelic expression of the X-linked gene MSN, pointing to a potential role in marsupial XCI, which warrants further investigation".

Row 203-204: The Xi-elect becomes depleted in histone modifications associated with active transcriptions at XCI onset in eutherians, but this to my knowledge hasn't been studied in marsupials.

We modified the sentence on line 170 to remove the kinetics aspect: "The Xi is generally depleted in histone modifications associated with active transcription and enriched in repressive histone modifications in both eutherians and marsupials (Chaumeil et al., 2011; Rens et al., 2010)."

Please consider citing Rens et al., 2010, PNAS article as an important study that profiled different acetylations and methylations in opossum X.

We have cited this paper in the revised version.

Row 261: Regarding the "no overlap" in XCI escapees, consider mentioning that this is in accordance with the major escapees being Y-linked gametologs with somatic expression. Since the murine and marsupial Y chromosomes diverged significantly (Bellot et al., Nature, 2014), it makes sense that the corresponding escapees will too.

We thank the Reviewer for pointing out this information and we agree with them. This is an interesting parameter that requires further exploration. For the sake of time, we decided not to include it in this Review. The sentence was therefore removed from the text.

Row 276: It's unlikely that the "absence of DNA methylation on the Xi contributes to a different pattern of XCI escape" because it is throughout the chromosome. Please see comment 11 about escapees.

We agree with the reviewer's comment and thus this sentence has been removed.

Row 293: Regarding iXCI in opossums being the only form of XCI in all tissues – consider mentioning that marsupial pluripotency is decoupled from XaXa status, which complements the paragraph's theme of iXCI

reversal in the mouse pluripotent inner cell mass. At the end of this / subsequent paragraph would be a good place (among others) to discuss theories and trends (see major comment above).

Please refer to our comment above.

Rows 320-321: ZGA and iXCI starts around 8 cell stage in opossum, and not as mentioned.

We have corrected the sentence on line 278 as such: "In mice, iXCI begins around the 2- to 4-cell stage, while in marsupials it initiates at the 8-cell stage, both coinciding with zygotic genome activation (ZGA)".

Row 325: References 79-81 appear in different format.

This was corrected.

Row 335: The statement that Xist may have "functions beyond XCI" due to the findings above, doesn't make sense. The accumulation of Xist/proteins/histone modifications in absence of transcriptional silencing, can be explained by the absence of a critical and unknown key component that makes all elements "work together", or by specially restricted accumulations of these elements (e.g intergenic area only) that cannot be further resolved by microscopy only. See again, major comment, hypothesis, thought.

This sentence was referring to the role of XIST in XCD discussed later in the Review but for clarity, we removed it.

Row 392: Define the different colours that Xa, Xd and Xi are represented with in this legend.

We modified the legend as suggested.

Row 403-404: I'm not sure I agree that a marsupial Xic-like region has not been clearly defined. The Xic is an X-inactivation centre; a site on the future inactive X from which the instructive silencing RNA is produced. Such a site has been demonstrated based on Rsx transgenic studies in mice. I wonder here whether the authors mean that a more complex Xic housing multiple ncRNAs akin to that in eutherians has not yet been defined in marsupials?

We modified this sentence on line 355 as such: "In contrast, the marsupial *Xic* remains poorly characterized, although emerging evidence points to the presence of antisense regulatory mechanisms reminiscent of those observed in mice (see below)."

Regarding the "mechnisms controlling RSX expression" mentioned in row 404, also applying for the statement in row 424: Please note the differential RSX promoter DNA methylation, which is maternally present and paternally absent, from Wang et al., 2014, Genome research, (which is cited in this review) as a likely source of RSX regulation. Also note that the Waters lab recently claimed that imprinted XCI is regulated by inheritance of a hypomethylated inactive X doi: 10.1073/pnas.2412185121

We have modified the paragraph on line 378 accordingly: "The mechanisms regulating RSX expression and iXCI in marsupials remain to be fully elucidated. Differential DNA methylation has been observed on the RSX promoter, with methylation occurring exclusively on the maternal active X. However, it remains to be determined whether this methylation influences RSX expression (Wang et al., 2014)."

Row 453: Please rephrase "which however revealed", this is not clear.

We added commas for better readability.

Dr. Claire Rougeulle
Insitutut Curie
26 rue d'Ulm
Paris, Paris 75005
France

Dear Claire,

I am pleased to inform you that your Review has been accepted for publication in EMBO reports. Your manuscript will be processed for publication by EMBO Press. It will be copy edited and you will receive page proofs prior to publication.

You will soon be contacted by Springer Nature to sign your publishing license. When you login to the customer service website, please use the following token to waive the article publication charges: MJEZMJM5ODQZNA. Should you experience any difficulty, please email publishing@embo.org.

If you have any questions, please do not hesitate to contact the Editorial Office. Thank you very much for your contribution to EMBO Reports!
